# Cudraxanthone D Ameliorates Psoriasis-like Skin Inflammation in an Imiquimod-Induced Mouse Model via Inhibiting the Inflammatory Signaling Pathways

**DOI:** 10.3390/molecules26196086

**Published:** 2021-10-08

**Authors:** Namkyung Kim, Soyoung Lee, Jinjoo Kang, Young-Ae Choi, Yong Hyun Jang, Gil-Saeng Jeong, Sang-Hyun Kim

**Affiliations:** 1Department of Pharmacology, Cell & Matrix Research Institute, School of Medicine, Kyungpook National University, Daegu 41944, Korea; nortonnklab@gmail.com (N.K.); jinjoo1kang@gmail.com (J.K.); korryy@hanmail.net (Y.-A.C.); 2Immunoregulatory Materials Research Center, Korea Research Institute of Bioscience and Biotechnology, Jeongeup 28116, Korea; sylee@kribb.re.kr; 3Departments of Dermatology, School of Medicine, Kyungpook National University, Daegu 41944, Korea; 4College of Pharmacy, Chungnam National University, Daejeon 34134, Korea

**Keywords:** *Cudrania tricuspidata* Bureau, cudraxanthone D, psoriasis, imiquimod, keratinocytes

## Abstract

Psoriasis is a chronic inflammatory skin disease accompanied by excessive keratinocyte proliferation. Corticosteroids, vitamin D3 analogs, and calcineurin inhibitors, which are used to treat psoriasis, have diverse adverse effects, whereas natural products are popular due to their high efficiency and relatively low toxicity. The roots of the *Cudrania tricuspidata* (*C. tricuspidata*) are known to have diverse pharmacological effects, among which the anti-inflammatory effect is reported as a potential therapeutic agent in skin cells. Nevertheless, its effectiveness against skin diseases, especially psoriasis, is not fully elucidated. Here, we investigated the effect of cudraxanthone D (CD), extracted from the roots the *C. tricuspidata* Bureau, on psoriasis using an imiquimod (IMQ)-induced mouse model and the tumor necrosis factor (TNF)-α/interferon (IFN)-γ-activated keratinocytes. IMQ was topically applied to the back skin of C57BL/6 mice for seven consecutive days, and the mice were orally administered with CD. This resulted in reduced psoriatic characteristics, such as the skin thickness and Psoriasis Area Severity Index score, and the infiltration of neutrophils in IMQ-induced skin. CD inhibited the serum levels of TNF-α, immunoglobulin G2a, and myeloperoxidase, and the expression of Th1/Th17 cells in splenocytes. In TNF-α/IFN-γ-activated keratinocytes, CD reduced the expressions of CCL17, IL-1β, IL-6, and IL-8 by inhibiting the phosphorylation of STAT1 and the nuclear translocation of NF-kB. Taken together, these results suggest that CD could be a potential drug candidate for the treatment of psoriasis.

## 1. Introduction

Psoriasis is a chronic inflammatory skin disease characterized by the abnormal interaction of keratinocytes with psoriasis-associated immune cells, such as neutrophils, dendritic cells, and activated T cells, in the epidermis [1,2]. Psoriatic lesions have a variety of clinical characteristics, such as scaling patches, redness, psoriasiform keratosis, and epidermal/dermal phenotype changes, including an increased skin thickness, and the infiltration of immune cells [3,4]. The accelerating skin inflammation in psoriasis is due to the expression of many cytokines and chemokines from the infiltrated immune cells (resident keratinocytes and neutrophils) [5]. Particularly, the tumor necrosis factor (TNF)-α and the interferon (IFN)-γ in keratinocytes accelerate the inflammatory response by stimulating the secretion and synthesis of various inflammatory mediators [6]. Therefore, it is important to control the secretion of the cytokines and chemokines involved in psoriasis.

Corticosteroids, calcineurin inhibitors, and vitamin D3 analogs are used as universal therapy for mild psoriasis in clinical practice [7]. These drugs can acutely reduce psoriatic characteristics, except that their long-term use may cause adverse effects such as skin atrophy, burning, pruritus, and edema [8,9,10]. Secukinumab and ixekizumab, which are selective IL-17 antagonists, have been approved by the U.S. Food and Drug Administration for moderate or severe psoriasis treatment in clinical practice. However, these are expensive and their use is limited in patients with inflammatory bowel disease, skin mucosal candidiasis, and infectious diseases [8,11,12]. Thus, natural ingredients and traditional medicines are more frequently studied due to the fewer side effects over the past several decades [13,14].

*Cudrania tricuspidata* is distributed throughout Korea, Japan, and China, and is widely used as the treatment for diverse disease, such as atopic dermatitis, mumps, insomnia, and acute arthritis [15,16,17]. *C. tricuspidata* contains many secondary metabolites, such as xanthones and flavonoids, which constitute the major possible active compounds. In addition, the root bark of *C. tricuspidata* is reported to exhibit anti-sclerotic, anti-inflammatory, monoamine oxidase suppressive effects [16,18]. In particular, xanthones and flavonoids are major components of *C. tricuspidate*. Cudraxanthone D (CD), extracted from the root of *C. tricuspidata* Bureau, is a natural xanthone with anti-inflammatory, neuroprotective, and antioxidant effects [16,19]. Despite the multiple pharmacological action of CD, its effects are unknown regarding skin diseases, especially psoriasis. Therefore, this study aimed to evaluate the effects of CD on an IMQ-induced mouse model and interpret the molecular mechanisms behind these anti-inflammatory effects.

## 2. Results

### 2.1. CD Reduced the Symptoms of Psoriasis-like Skin Inflammation

In a previous study, our research group succeeded in isolating pure cudratricusxanthone A (CA) and CD from the roots of *C. tricuspidata* Bureau [20]. The chemical structures of the two compounds are shown in Appendix A. We conducted a drug screening of CA and CD regarding their anti-inflammatory effects in keratinocytes after cytotoxicity.

To assess whether the compounds of the two chemicals exhibited a cytotoxicity in HaCaT cells, we performed a 3-(4,5-Dimethylthiazol-2-yl)-2,5-Diphenyltetrazolium Bromide (MTT) assay with results available up to 1 μM (Appendix A). Then, we first compared the anti-inflammatory effects of CA and CD in TNF-α/IFN-γ-activated keratinocytes and found that CD, but not CA, reduced the expression of proinflammatory cytokines (IL-1β and IL-6) (Appendix A). Therefore, further studies should focus on CD.

To evaluate the effects of CD on psoriasis-like skin inflammation, we applied it to the mouse skin using imiquimod (IMQ) cream [21]. An experimental schedule for this is shown in Appendix A. The repeated application of IMQ dramatically increased its psoriatic characteristics, including skin thickness, scaling, erythema, and dry skin. During the experimental period, IMQ-induced skin showed a significant increase in skin thickness, whereas the oral administration of CD reduced skin thickness (Figure 1a). The current gold standard for the assessment of extensive psoriasis is the Psoriasis Area Severity Index (PASI) score, which assesses scaling, erythema, and thickness [22]. Thus, we measured the PASI scores daily to verify the efficacy of CD on IMQ-induced mouse skin. IMQ-induced psoriasis skin increased the PASI score in all three aspects of scaling, erythema, and thickness in mouse skin. The oral administration of CD diminished the PASI scores in a dose-dependent manner (Figure 1b). The bodyweights of the mice did not significantly change during the experimental period, and no toxicity due to CD was noted (Figure 1c). A histological analysis revealed that CD alleviated the psoriatic characteristics, such as epidermal hyperplasia, parakeratosis, and immune cell infiltration, as well as reducing the thickness of the epidermis and dermis (Figure 1d,e).

### 2.2. CD Reduced the Mediators of Psoriasis-Related Inflammation in an IMQ-Induced Mouse Model

The acceleration of skin inflammation in psoriasis occurs through the interaction of diverse cytokines and chemokines produced by immune cells, such as neutrophils and resident keratinocytes [5]. Therefore, we performed a quantitative polymerase chain reaction (qPCR) in IMQ-induced mouse skin to investigate the inhibitory effect of CD on inflammatory-related gene expression. The continuous IMQ-application upregulated the expression of inflammatory-related genes, such as C-X-C Motif Chemokine Ligand (CXCL) 1, interleukin (IL)-25, IL-17A, IL-6, IL-1β, IL-36, IL-4, CD3, and IFN-γ. However, CD inhibited the expression of these genes except IL-4 (Figure 2). 

The topical application of IMQ was shown to induce psoriasis-like skin inflammation through the activation of the Toll-like receptor (TLR)7 [23]. Therefore, we examined TLR7 expression to determine whether the relief of psoriatic characteristics by the oral administration of CD was caused by inhibiting TLR7. As a result, CD did not reduce the expression of TLR7 (Appendix A).

To investigate the suppressive effect of CD on the psoriatic sera, we measured the levels of TNF-α by using an enzyme-linked immunosorbent assay (ELISA), immunoglobulin (Ig)G2a, and myeloperoxidase (MPO). In general, the serum levels of the psoriasis patients had increased levels of TNF-α and IgG2a [24,25]. In our study, the IMQ-induced mouse sera increased TNF-α and IgG2a levels, but these levels were reduced via oral administration (Figure 3a). In particular, MPO was associated with CD4^+^ T cells and neutrophils in the skin [26]. Specifically, MPO was a marker of neutrophils, and a typical histopathological hallmark of psoriasis was the abundance of the neutrophils in the psoriatic skin lesions [27]. Therefore, we assessed the serum MPO levels, observed MPO expression using immunohistochemistry (for MPO) in IMQ-induced skin, and found that they both were decreased by the oral administration of CD (Figure 3b,c). To assess the actions of CD on the neutrophil-related gene expression in IMQ-induced skin, we examined the expression of the neutrophil markers, CD11b and the Lymphocyte antigen 6 complex locus G6D (Ly6G) [28,29]. In comparison to the IMQ-induced mice group, the oral administration of CD decreased the expression levels of CD11b and Ly6G (Figure 3d).

### 2.3. CD Reduced the Inflammation-Related Gene Expression by Inhibiting NF-κB and STAT1 in Activated Keratinocytes

To investigate the anti-inflammatory effect of CD, we measured the inflammation-related gene expression and secreted proteins using qPCR and ELISA in TNF-α/IFN-γ-activated keratinocytes. We found that CD reduced the expression of CCL17, IL-6, IL-8, and IL-1β in keratinocytes (Figure 4a). Additionally, TNF-α/IFN-γ increased the secretion of IL-6 and CCL17 in keratinocytes, but these were also reduced by CD (Figure 4b).

The activity of the transcription factors responsible for the expression of the inflammation-related genes were investigated to understand the signaling mechanism for the anti-inflammatory effect of CD in activated keratinocytes. As shown in Figure 4c, TNF-α/IFN-γ were shown to induce the phosphorylation of STAT1 and the nuclear translocation of NF-κB. However, CD inhibited STAT1 phosphorylation, IκBα degradation, and NF-κB translocation in the activated keratinocytes (Figure 4c). STAT1 and NF-κB activation were also confirmed in the IMQ-induced psoriasis model to define the biological activity of CD. As shown in Appendix A, IMQ activated STAT1 and NF-κB in mouse skin, but this was inhibited by CD.

## 3. Discussion

The incidence of psoriasis has steadily increased due to a variety of harmful factors, such as air pollution, smoking, mechanical stress, dyslipidemia, and hypertension [30,31]. There are various treatment options for psoriasis, but topical corticosteroids are mainly used due to their immunosuppressive, anti-inflammatory, and antiproliferative effects. However, the long-term use of these drugs also has limitations because of their side effects, such as atrophy, hyperpigmentation, hypercoagulability, and dyslipidemia [32]. Therefore, this study aimed to find a suitable therapeutic candidate derived from natural compounds with a proven efficacy in treating this skin disease. The diverse herbal medicinal products extracted from dietary plants were previously studied, especially for the treatment of skin diseases [33]. *C. tricuspidata* has long been used in East Asia for the treatment of rheumatism, hepatitis, jaundice, and gonorrhea. Additionally, the compounds derived from *C. tricuspidata* are reported to have anti-inflammatory, antioxidant, hepatoprotective, anti-obesity, neuroprotective, and skin-protecting effects with a biological activity [34,35,36,37]. Particularly, CD extracted from the roots of *C. tricuspidata* demonstrates anti-cancer effects in an oral squamous cell carcinoma, as well as anti-inflammatory and antioxidant effects [38,39]. Our study specifically aimed to define the role of CD in IMQ-induced psoriatic skin. 

The IMQ-induced mouse model closely demonstrated the significant psoriatic characteristics, including psoriatic pathological changes, skin erythema, thickening, scaling, and inflammatory infiltrate, consisting of T cells and neutrophils, and a Th cell-mediated immune response [40]. In comparison, the most common clinical characteristics of psoriasis include keratinocyte hyperproliferation, increased angiogenesis, and skin inflammation due to the elevated inflammatory cytokines associated with Th cells (Th1, Th17, and Th22) in psoriatic lesions [41,42,43]. Consequently, we showed that the skin of IMQ-induced mice mimics the psoriatic characteristics with increased infiltration of inflammatory cells and increased levels of psoriasis-related cytokines and chemokines such as CXCL1, IL-25, IL-17A, IL-6, IL-1β, IL-36, CD4, and IFN-γ. However, CD exhibited a suppressive effect on these psoriatic characteristics and decreased the inflammation-related cytokine and chemokine in IMQ-induced skin. Thus, it can be said that CD alleviated the inflammatory response by decreasing the immune cell-related gene expression involved in the development of psoriasis. In addition, CD decreased the gene expression of CD3 but not IL-4. From these results, we speculate that CD influenced the infiltration of T cells.

To investigate whether CD inhibited TLR7 expression in the IMQ-induced mice, we measured the amount of TLR7 expression. As a result, CD did not suppress the expression of TLR7. Thus, we speculate that the alleviation of the psoriatic characteristics by CD might cause an inhibitory effect through the inhibition of TLR7 downstream signals or other pathways.

Neutrophils, because of their pathogenic roles, are also related to chronic inflammation and autoimmune diseases in psoriasis [26]. In line with this, the neutrophil elastase inhibitors are shown to treat psoriasis in a mouse model of the disease [44]. Therefore, we assessed the expression of MPO in the serum and skin tissue, as well as the serum levels of cytokines and Ig. CD was found to decrease the serum levels of psoriasis-related inflammation mediators (i.e., TNF-α, IgG2a, and MPO). The oral administration of CD was also able to decrease the MPO expression in IMQ-induced skin. Furthermore, we verified neutrophil markers (CD11b and Ly6G) to determine whether the CD decreases neutrophil infiltration or reduces the transcriptional changes in existing cells. The oral administration of CD decreased the expression of CD11b and Ly6G. Therefore, we suggest that CD exerts an anti-inflammatory effect by reducing the developmental factors in psoriasis, especially neutrophils.

Several studies reported that the activated keratinocytes secreted proinflammatory cytokines, which accelerated the development of psoriasis [45]. Keratinocytes are the main sources of gene expression that accelerate the progress of psoriasis [6] because the enhanced production of TNF-α and IFN-γ promotes inflammation [46]. Therefore, we investigated the effects of CD on keratinocytes stimulated with TNF-α and IFN-γ. Alternatively, the excessive activation of the STAT1 and NF-κB signaling pathways aggravated skin inflammation in psoriasis [47,48]. We observed that CD suppressed the gene expressions of CCL17, IL-6, IL-8, and IL-1β, as well as the secretion of IL-6 and CCL17, through the inhibition of transcription factors, STAT1 and NF-κB, in TNF-α/IFN-γ-activated keratinocytes. 

In this study, the oral administration of CD alleviated the characteristics of psoriasis in the IMQ-induced mice model and CD decreased the activation of keratinocytes via the inhibition of pro-inflammatory cytokine production by suppressing the activity of STAT1 and the NF-κB pathways. Based on these data, we suggest that CD could be a potential therapeutic agent for psoriasis.

## 4. Materials and Methods

### 4.1. Reagents, Cell Maintenance, and Cell Viability

All reagents were purchased from Sigma-Aldrich (St. Louis, MO, USA) unless otherwise noted. To use the activation of keratinocytes, recombinant human proteins, TNF-α and IFN-γ (R&D systems, Minneapolis, MN, USA), were used. For the cell experiment, CA and CD were dissolved with dimethyl sulfoxide (DMSO). For the animal experiments, CD was dissolved with phosphate-buffered saline (PBS). For IMQ cream (5% Aldara™), 62.5 mg was used (DONG-A pharmaceuticals, Seoul, Korea). CD and Dexa were administrated orally by gavage for 7 consecutive days.

Keratinocyte cells (HaCaT cell line) were maintained in Dulbecco’s modified Eagle’s medium (Gibco, Grand Island, NY, USA) containing 10% fetal bovine serum (Gibco) and antibiotics (100 U/mL penicillin G, 100 μg/mL streptomycin, Gibco). Cells were cultured at 37 °C in 5% CO_2_. 

Cell viability was determined using the MTT assay. Briefly, HaCaT cells (1 × 10^4^ cells/well in a 96 well plate, *n* = 5) were treated with CA or CD at different concentrations (0–100 μM) for 24 h, followed by incubation with 20 μL of MTT solution (5 mg/mL) for 4 h. The formed formazan crystals were dissolved in 100 μL of DMSO (99.5%). Absorbance was determined using a plate reader at 570 nm (Molecular Devices, Sunnyvale, CA, USA). The relative cell viability is presented as a percentage compared to non-treated control cells (100%).

### 4.2. Plant Materials and Isolation of CD

The dried roots of *Cudrania tricuspidata* Bureau were purchased from the Yangnyeong herbal medicine market (Daegu, Korea), and were identified by Jeong of the College of Pharmacy, Keimyung University, Daegu, Republic of Korea. A voucher specimen (KMU-2017-05-11) of the plant was deposited.

The dried roots of *C. tricuspidata* Bureau (12 kg) were extracted with EtOH at 60 °C for 3 h, then evaporated under reduced pressure. The dried EtOH (463.4 g) extract was suspended with H_2_O, and the resulting H_2_O layer was partitioned three times with hexane (54.4 g), CH_2_Cl_2_ (72.4 g), and EtOAc (151.1 g). The CH_2_Cl_2_-soluble fraction was loaded onto a silica column (8 × 60 cm, silica-gel 70–230 mesh), eluted with hexane-EtOAc (gradient from 100:0 to 0:100), then with EtOAc-MeOH (gradient from 100:0 to 0:100). The eluates were combined for thin-layer chromatography analysis, providing 11 fractions (FR-1–FR-11). Among them, FR-6 (149.7 mg) was purified via prep-high performance liquid chromatography with CH_3_CN:H_2_O (from 50:50 to 100:0) to obtain CA (12.4 mg) and CD (51.2 mg). According to Park et al., CA and CD from roots of C. tricuspidata can be identified by comparing the values of spectroscopy data [20]. As a results, these extracts consisted of xanthone compounds, and the quantitative contents of CA and CD were confirmed as 0.46 ± 0.02% and 1.53 ± 0.06%, respectively [20].

### 4.3. Ethics Statement

C57BL/6J mice (8-week-old, female, *n*=35, 5 mice/cage) were purchased from DBL Co., LTd. (Daejeon, Republic of Korea). All mice were housed per cage in a laminar air flow room maintained at 22 ± 2 °C, with a relative humidity of 55 ± 5% and a 12 h light/dark cycle for the experimental period. The care and treatment of the mice were in accordance with the guidelines established by the Public Health Service Policy on the Humane Care and Use of Laboratory Animals. All experiments were approved by the Institutional Animal Care and Use Committee of Kyungpook National University (KNU-2020-0022).

### 4.4. Observation in the Back Skin of Mouse and Sample Collection

After 24 h of IMQ application, skin thickness was measured using a 7301-dial thickness gauge (Mitutoyo, Co., Tokyo, Japan) and was assessed using the PASI. The PASI independently scores scaling, erythema, and thickness from 0 to 4 as follows: 0: none, 1: slight, 2: moderate, 3: marked, and 4: very marked.

All mice were euthanized with CO_2_ gas, after which the back skin was collected. The back skin was used for histological analysis and qPCR. Whole blood was collected from the abdominal vena cava, and the serum was isolated by centrifuging at 400× *g* for 15 min at 4 °C for ELISA.

### 4.5. ELISA

IgG2a (BD Biosciences, Oxford, UK), TNF-α (BD Biosciences), MPO (R&D System Inc., Minneapolis, MN), IL-6 (R&D System Inc.), and CCL17 (R&D System Inc.) were measured using a specific ELISA kit according to the manufacturer’s protocol. Absorbance was measured at 450 nm using a spectrophotometer (VersaMax™ Microplate Reader, Molecular Devices, San Jose, CA, USA). The calculation and analyses of data were conducted using the SoftMax Pro software version 6 (Molecular Devices). IgG2a, TNF-α, and MPO were measured in IMQ-induced mouse sera. On the other hand, HaCaT cells (2 × 10^5^ cells/24-well plate) were pretreated with CD (0.01, 0.1, or 1 μM) or Dexa (1 μM) for 1 h, and then stimulated with TNF-α (10 ng/mL) and IFN-γ (10 ng/mL) for 15 h. Then, the collected supernatant was centrifuged at 2500× *g* and 4 °C for 5 min in order to detect secretory cytokines (IL-6) and chemokines (CCL17).

### 4.6. qPCR

RNA was isolated using the RNAiso Plus Kit (Takara Bio Inc., Shiga, Japan) protocol. HaCaT cells (2 × 10^5^ cells/well in a 24-well plate) were pretreated with CD (0.01, 0.1, or 1 µM) or Dexa (1 µM) for 1 h and stimulated with TNF-α (10 ng/mL) and IFN-γ (10 ng/mL) for 6 h. On the other hand, the back skin tissues were homogenized using the TissueLyser II (Qiagen, Hilden, Germany). The synthesis of cDNA was completed using the RevertAid RT kit (ThermoFisher scientific, Waltham, MA) according to the manufacturer’s protocol, whereas qPCR was performed using the QGBlue PCR 2X Master Mix (Cellsafe, Yongin, Republic of Korea). The primer sequences for PCR are described in Appendix A. mRNA expression was normalized with glyceraldehyde 3-phosphate dehydrogenase, both in vivo and in vitro. Relative quantification using the double delta Ct method was performed with StepOnePlus PCR system software (ThermoFisher scientific) following the manufacturer’s instructions.

### 4.7. Histological and Immunohistochemistry Analysis

The back skin tissues were fixed with 10% formaldehyde, embedded in paraffin, then sliced into 6-µm sections. The tissue section slides were stained with hematoxylin and eosin (H&E) and MPO staining for immunohistochemistry (IHC). For H&E staining, sectioned tissues were deparaffinized by xylene and rehydrated by graded alcohol series. Then, tissues were preferentially stained with Harris’ hematoxylin solution for 6 min at room temperature and rinsed in tap water until the water was colorless. Subsequently, we soaked the tissues in saturated lithium carbonate solution for 2 min then rinsed them with tap water. Staining was finally performed with eosin Y ethanol solution for 40 s. Next, we proceeded to rinse, dehydrate, clear, and mount according to standard protocol. For IHC, the tissue sections were deparaffinized in xylene and rehydrated in decreasing concentrations of ethanol. Antigen retrieval was performed in 10 mM citrate buffer (pH 6.0) for 30 min in a microwave, then placed in 3% hydrogen peroxide in methanol for 5 min. The following primary antibody was used with MPO (1:100; Abcam, Cambridge, UK). Incubation in horseradish peroxidase-conjugated secondary antibody (Cell signaling, MA, Danvers) was subsequently performed, followed by development with diaminobenzidine (Vector Laboratories, Burlingame, CA, USA) and counterstaining with hematoxylin (Vector Laboratories). The H&E- and MPO-stained slides were observed using a Carl Zeiss microscope (Jena, Germany). Epidermal and dermal thickening was measured using a stage micrometer (Carl Zeiss), and MPO-associated cells were observed at 200× magnification. 

### 4.8. Western Blot

At the end of animal experiments, the back skin was gathered in 300 µL of RIPA buffer (Biosesang, Seongnam, Korea), containing a protease/phosphatase inhibitor cocktail (Roche, Mannheim, Germany). Afterwards, skin tissues were homogenized using the TissueLyser II (Qiagen), then the tissue lysates were centrifuged at 1200× *g* for 30 min at 4 °C, and the supernatant was collected. On the other hand, HaCaT cells (1 × 10^6^ cells/well in a 6-well plate) were seeded and pretreated with CD (1 µM) or Dexa (1 µM) for 1 h, then treated with TNF-α (10 ng/mL) and IFN-γ (10 ng/mL) for 15 min_._ After stimulation, cells were washed twice in 1 mL of cold PBS with 100 nm Na_3_VO_4_, and the cell lysate was gathered in 150 µL of lysis buffer (containing 150 mm NaCl, 10 mm HEPES, 2 mm MgCl_2_, 1 mm EDTA, 10 mm KCl, 0.5 mm DTT, 0.5 mm PMSF, and 0.5% Triton X-100; pH 7.9). The cell lysates were centrifuged at 1200× *g* for 30 min at 4 °C, and the supernatant was collected as cytosol proteins. The pellets were washed with PBS and lysed in 40 µL of ice-cold RIPA buffer (Biosesang), sonicated, then centrifuged at 15,000× g for 20 min at 4 °C. The supernatants were collected as nuclear proteins. All lysis buffers were contained in protease/phosphatase inhibitor cocktail (Roche). All proteins were quantified via the Bradford (Bio-rad laboratories, Hercules, CA, USA) method and loaded with 10% sodium dodecyl sulfate-polyacrylamide gel electrophoresis, then transferred into nitrocellulose membranes (Pall Corporation, Ann Arbor, MI, USA). Next, membranes were observed via Ponceau S (Bio-rad laboratories) staining and with blocking solution containing 3% bovine serum albumin in Tris-buffered saline with 0.1% Tween 20. The membranes were incubated with specific primary antibodies (Appendix A). E2-Western blot stripping buffer (DoGenBio, Seoul, Korea) was used for the removal of primary and secondary antibodies from a single membrane for reprobing. Immunodetection was performed using SuperSignal West Pico chemiluminescent substrate (Thermo Scientific, Waltham, MA, USA) by G:Box Chemi XRQ (SYNGENE, Cambridge, UK).

### 4.9. Statistical Analysis

Statistical analyses were carried out using the Prism 7 GraphPad software (San Diego, CA, USA). Treatment effects were analyzed via one-way analysis of variance followed by Dunnett’s test. Results are expressed as mean ± standard error of the mean (SEM). A *p* < 0.05 was considered statistically significant.

## 5. Conclusions

The oral administration of CD alleviates the psoriatic characteristics by reducing the inflammatory mediators in skin inflammation on IMQ-induced psoriasis mouse. CD exerts an inhibitory effect against the activation of STAT1 and NF-κB. Based on these findings, natural compounds such as CD could be used as a treatment for psoriasis.

## Figures and Tables

**Figure 1 molecules-26-06086-f001:**
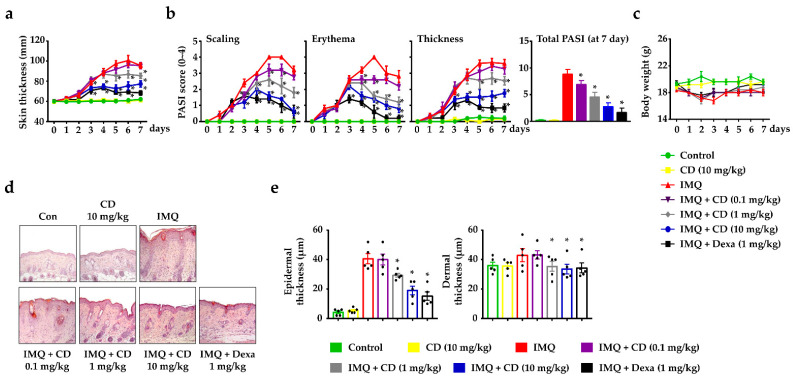
Effects of cudraxanthone D (CD) on psoriasis-like skin inflammation. (**a**) Skin thickness of mice measured 24 h after imiquimod (IMQ) or drug (CD or Dexa) application with a dial thickness gauge No. 7301. (**b**) Psoriasis Area Severity Index (PASI) accounting for scaling, erythema, and thickness were scored from 0 to 4 in IMQ-induced skin inflammation, and total PASI score at day 7. (**c**) Body weight during the experimental period was measured using a HKC65050 electronic balance for 7 consecutive days. (**d**) Representative micrographs of the skin were stained with hematoxylin and eosin (H&E). Magnification: 200×, scale bar: 50 μm. (**e**) Epidermal and dermal thickness. At 200× magnification, the epidermal and dermal thickening was analyzed with a stage micrometer. Each data point represents the mean ± SEM of two independent samples. * *p* < 0.05 compared to the IMQ-induced group only. IMQ: imiquimod, CD: cudraxanthone D, Dexa: dexamethasone.

**Figure 2 molecules-26-06086-f002:**
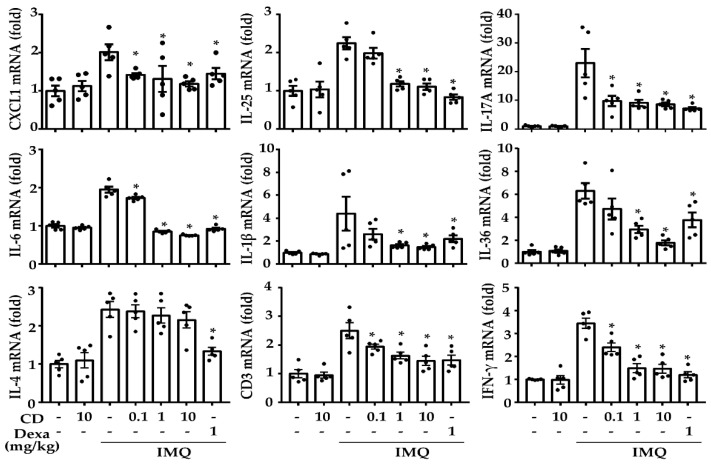
Effects of CD on psoriasis-associated cytokines and chemokines in IMQ-induced skin. The mRNA expressions of IMQ-induced skin lesions were measured via quantitative polymerase chain reactions (qPCR) and normalized with GAPDH. Each data point represents the mean ± SEM of two independent samples. * *p* < 0.05 compared to the IMQ-induced group only. IMQ: imiquimod, CD: cudraxanthone D, Dexa: dexamethasone.

**Figure 3 molecules-26-06086-f003:**
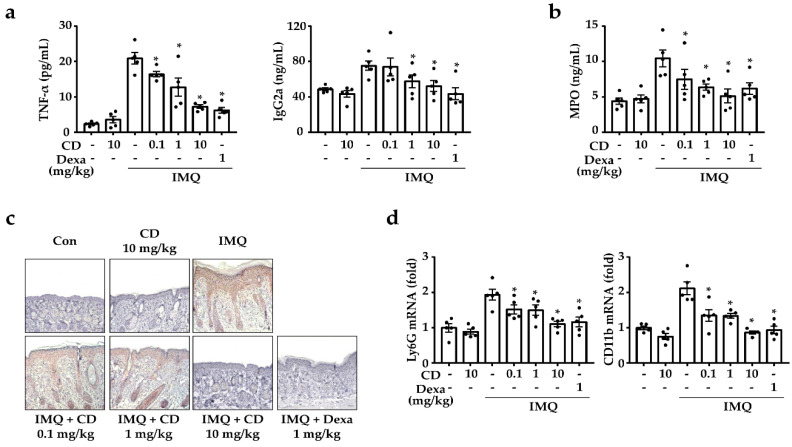
Effects of CD on serum immunoglobulin levels and infiltration of MPO-associated cells. After the in vivo experiments, the mice were sacrificed, after which their blood and skin were harvested. (**a**,**b**) The serum levels of tumor necrosis factor (TNF)-α, IgG2a, and MPO were measured via sandwich enzyme-linked immunosorbent assay (ELISA). Each data point represents the mean ± SEM of the two independent samples. * *p* < 0.05 compared to the IMQ-induced group only. (**c**) Tissue slides stained with immunohistochemistry (IHC) (for myeloperoxidase [MPO]). (**d**) The expressions of neutrophil markers in IMQ-induced skin lesions were measured by qPCR. Magnification: 200×, scale bar: 50 μm. MPO: myeloperoxidase, IMQ: imiquimod, CD: cudraxanthone D, Dexa: dexamethasone.

**Figure 4 molecules-26-06086-f004:**
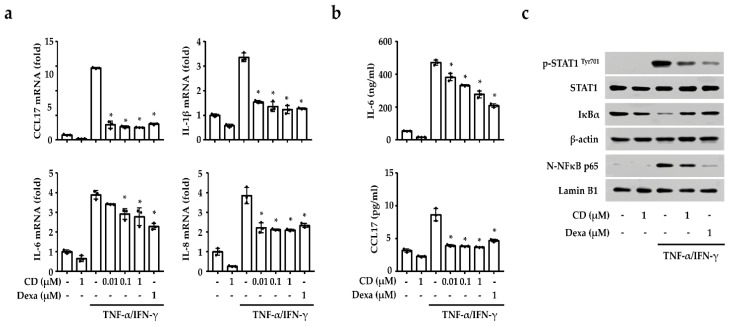
Effects of CD on the mRNA expression and secreted protein in activated keratinocytes with TNF-α/IFN-γ. (**a**) The expression levels of CCL17, IL-1β, IL-6, and IL-8 measured via qPCR. (**b**) Effects of CD on levels of secretory cytokine and chemokine measured by ELISA. (**c**) Translocation of NF-κB and phosphorylating of STAT1 were detected via Western blotting. The STAT1, β-actin, and lamin B1 bands were used as loading control. Each data point represents the mean ± SEM of three independent samples. * *p* < 0.05 compared with TNF-α/IFN-γ-stimulated group only. CD: cudraxanthone D, Dexa: dexamethasone.

## Data Availability

Not applicable.

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
