# Peer review of "Cudraxanthone D Ameliorates Psoriasis-like Skin Inflammation in an Imiquimod-Induced Mouse Model via Inhibiting the Inflammatory Signaling Pathways"

_molecules, 2021, doi:10.3390/molecules26196086_

Round 1

Reviewer 1 Report

The manuscript "Cudraxanthone D ameliorates psoriasis-like skin inflammation in an imiquimod-induced mouse model via inhibition of STAT1 and NF-κB activation" is an engaging article and some suggestions are presented below to improve its readability and cleareness.

Abstract

Line 30: The syntaxis of this line can be improved. Besides, authors should pay attention to how they describe their findings. For example, is CD acting on the disease or just ameliorating symptoms? In my opinion, something described as a supplement (as described before authors) is (as themselves said) a relief for symptoms or an adjuvant for drug therapy.

 Introduction

Line 38: I suggest changing the word “scaling” by another word, like “scaling patches” will make the sentence more clear

 Line 41: I suggest changing the word “various” by “many,” this word is not incorrect, but it is repeated in line 44, so it looks redundant.

Line 44: This phrase can be improved as follows “It is important to control cytokines and chemokines secretion…” and “of psoriasis

Line 48: This is an unfair asseveration about the adverse effects of topical psoriasis treatment. These treatments are usually well-indicated by health care providers to avoid this kind of negative effect. On the one hand, according to the Cochrane review made by Mason et al. (2013), most of the considered studies did not report significant local or systemic effects. On the other hand, according to Soleymani (2015), cancer advertising is related to UV phototherapy. These details must be specified. Also, the references used by the authors are relatively old (more than five years). Newer review papers, including information about adverse events in drugs for psoriasis, have been published since then. Armstrong and Read have published a recent brilliant review on this matter (JAMA. 2020;323(19):1945-1960. doi:10.1001/jama.2020.4006).

Lines 46 and 49: These treatments belong to different therapy schemes. The first group is for mild psoriasis, and the second is indicated for moderate to severe psoriasis. I suggest you briefly explain this difference between therapies used in psoriasis in clinical practice.

Line 49: Could you please change “sekunumab” to “secukinumab”? This is the correct name of the recent FDA approved drug for psoriasis treatment fabricated by NOVARTIS

Line 54: It is essential not to generalize on this matter. First, there are many natural products considered for the treatment of many diseases in different regions around the world. Second, most of them are not well characterized on their side effects, partly because research focuses on pharmacological rather than toxicological activity. On the other hand, when authors refer to traditional medicine, they need to specify which one?... traditional medicine from what region?

Line 58:  change the word “various” by a synonym as it its “a variety of” or “diverse”… diseases.

Line 61: contains “many secondary metabolites” such as xanthones and flavonoids “that constitutes the major possible active compounds”

Results

Line 72: Authors should avoid using 1st persons (singular or plural, e.g., “we”) as far as possible. I suggest: “In a previous study”… “our research group succeeded in…”

Line 77: I suggest adding the word “compounds” after “the two chemicals…”. On the other hand, Figures S1 b, c, and d: please, improve figures quality. The resolution in these 3 figures is not enough for an evident appreciation.

Line 80: I would use “Therefore” instead of “thus”

Discussion

Line 209: authors described the role of Th1 and Th17 cells in psoriasis pathogenesis, but they did not mention anything about Th22, which plays an essential role in the disease process. This is an excellent opportunity to introduce it briefly.

Line 243: It is essential to make clear that anti-inflammatory activity observed in this study is from in-vivo and in-vitro models

Materials and methods

Line 273: “Thrice” is not incorrect and is formal, but I would prefer to use “three times”

Line 280: The reference cited by the authors here [20] is a paper written in Korean by the same or related research group. Enough data is provided in English in tables and chromatograms figures, but the revisor cannot correctly interpret the text.

Line 346: Please, when referring to microscope magnification, use capital “X” to assign the magnification.

Line 370: change “by” for “with”

Line 373: “Bradford” method is the person's last name who described such a method for protein quantification. Last names have a capital letter only at the first letter.

Please also check for plagiarism. We found some issues,.

Reviewer 2 Report

Review of Kim et al 2021

The study by Kim et al demonstrates the capacity of cudraxanthone D, a natural extract, to negate the local inflammatory skin response in a mouse model of psoriasis. The manuscript is well structured and presents interesting findings to the field, and believe that additional modifications can strengthen the findings of the paper:

  1. The results demonstrating the splenomegaly in the untreated psoriasis model is interesting. Could the authors expand on what populations of cells are enhanced in this case, and what differences are observed with CD treated mice.
  2. The FACS analysis performed in Figure 4b is not convincing and appears to be a technical flaw (eg. Compensation issue, fluorescent aggregates etc). Indeed, IL-17 and IFNg production would not be expected to be high without stimulation, so the experimental design could be improved by performing a PMA/Ionomycin/BFA stimulation on the spleens. It would also raise the significance of the study if this stimulation and detection of cytokines could be performed in the psoriatic skin.
  3. A powerful, albeit bonus, experiment to confirm the mechanistic action of CD-mediated STAT1/NF-kB activation would be to sort-purify the keratinocytes of control, CD-treated or DEX-treated mice and assess pSTAT1 and NF-kB translocation via WB. However, I recognise that this is a technically challenging experiment, and I appreciate that the authors have not overstated the current results therefore am satisfied with this discussion as it stands.

Additional comments:

  • Could the authors include the number of independent experimental repeats performed? If the experiment was only performed once, an additional repeat would be required to validate the results
  • For transparency, it’s encouraged to show individual data points on the bar graphs
  • Could the authors expand upon the route of administration in the methods section? In the main text, it was mentioned that CD was administered orally but this was not clarified in the methods. Was this via gavage? Oral feeding?
  • The description of Flow Cytometry Analysis could be improved, in particular the antibodies and flourochromes used is unclear. Please include the clones of the antibodies too.

Reviewer 3 Report

Major Points

1. The overall rationale for the experiments is weak. The logic behind the cytokines and chemokines chosen seems arbitrary, and the authors do not connect their findings back to the mechanisms underlying psoriasis or their proposed hypothesis of inhibiting proliferation.

2. The authors need to more rigorously test proliferation (as opposed to cytokine production, etc) if that is their stated hypothesis. Alternatively, if they are testing a different hypothesis, that should be stated instead.

3. Based on the data presented, it is unclear if the authors propose CD acts by preventing initiation of psoriasis (is it a TLR7 inhibitor?), or by preventing disease progression. This distinction would suggest more clearly focused experiments, as would specifically testing proliferation.

4. The proliferation angle seems falsified by the MTT findings and use of a dose that does not impair cellular survival.

5. The authors need to determine if the changes in qPCR results are due to infiltration of new cells or changes in transcription for existing cells. For example, a neutrophil marker needs to be analysed to determine if fewer neutrophils contributed to the cDNA sample.

6. Along with Th1 and Th17 cytokines, Th2 cytokines (and general T cell markers) should be examined to determine if the changes are due to a shift in T cell polarization, changes in T cell infiltration, or transcriptional activity of the existing cells.

7. CD seems to generally suppress all responses observed. A control cytokine that is unaffected by CD is needed to help determine specificity. In a similar vein, does CD broadly block TLR signailng?

8. In Figure 4b, the data do not support the conclusions made. The authors do not have any IL17+ or IFNg+ T cells, much less double positive cells. The double positive events are autofluorescence. Similarly, the SSC x CD4 includes many non-T cells based on scatter. This figure either needs to be removed (along with attendant conclusions like lines 227-229), or repeated with correct gating. To get better IL17/IFNg stimulation, the authors might consider stimulating splenocytes ex vivo with anti-CD3/CD28 to improve T cell responses.

9. Flow cytometry details need clarification. Lines 352-353, ‘PerCP-Cy5.5-FITC’ and other listed dye conjugates do not exist. Please double check which one is supposed to be PerCP-Cy5.5 and which others are FITC, PE, etc. The number of events analyzed (ie CD4+ cells) needs to be listed. Was compensation performed? If so, was it done on the machine or during analysis? Please list the amount of antibody and number of cells stained; most manufacturer protocols recommend titration prior to use, or provide a range. Instead of antibody kits, individual clones should be listed. PMCID: PMC2773297 is one resource to consider in describing flow cytometry.

10. Figure S2 needs total Stat1 and total NF-kB blots. Preferably, the existing blots should be stripped and reprobed.

11. In Table S1, the antibody dilution used should be listed. Secondary antibodies should also be described.

12. The staining times for histology seem unnecessarily long. How do the results change if a more typical staining protocol with shorter staining duration is used?

Minor Points

1. Rationale in lines 21-22 in abstract is weak.

2. Minor grammar changes needed, eg line 91 ‘mouse’ instead of ‘mice’, line 310 and elsewhere: ‘Alternatively’ instead of ‘Contrarily’

3. Fig 1 would be strengthened by adding a bar chart showing maximal PASI score.

4. Line 365, were both TNF and IFN used at 10 ng/mL?

5. Method of qPCR analysis should be described. Was delta(delta(Ct)) used?

Round 2

Reviewer 1 Report

Thanks for taking into account the suggestions made, the manuscript now is more engaging and address findings within the topic presented. I only suggest to change secukikunumab (line 59) to secukinumab.

Reviewer 2 Report

I am satisfied with the adjustments and find the manuscript suitable for publication. 

Reviewer 3 Report

The authors addressed some of my concerns. A few key points remain outstanding:

Major Points

1. The authors’ data do not discriminate between Stat/NF-KB reduction due to less inflammation overall, or if CD is specifically targeting those proteins. Therefore, either additional data are needed to substantiate these conclusions, or the title and discussion at line 321 need to be changed.

2. The CD11b/Ly6G mRNA data presented in the response letter should be in the manuscript.

3. Data provided on TLR7 do not address the issue of initiation vs progression. The amount of TLR7 mRNA is secondary to its ability to signal. This issue should be discussed in the manuscript, along with caveats on specificity/target of CD.

4. Line 297, it is not correct to conclude CD decreased activation. It is more likely that cells were not recruited in the first place.

5. Line 298, it is not correct to say that CD ‘inhibited CD3’. Reduction in mRNA is not inhibition.
